# TIME to Change: An Evaluation of Practical Action Nepal's Results Based Finance Program

**Benjamin L. Robinson** [1,*], **Mike J. Clifford** [1] and **Sarah Jewitt** [2]

1   Faculty of Engineering, University of Nottingham, University Park, Nottingham NG7 2RD, UK; mike.clifford@nottingham.ac.uk
2   School of Geography, University of Nottingham, University Park, Nottingham NG7 2RD, UK; sarah.jewitt@nottingham.ac.uk
*   Correspondence: benjamin.robinson@nottingham.ac.uk

**Abstract:** Set against the United Nations Sustainable Development Goal 7, and the need to increase biomass Improved Cookstove (ICS) adoption and sustained use across the globe, this paper presents an evaluation of Practical Action Nepal's (PAN) Results Based Financing for Improved Cookstove Market Development in Nepal (RBF) project, which was conducted between January and April 2020. Nepal has a long history of International Development assistance, yet 65.8% of rural households still use firewood as their primary source of energy. With this in mind we aimed to understand the barriers, enablers and engagement strategies for the adoption and sustained use of Improved Cookstoves (ICS), map key stakeholder role perceptions and interactions, and identify areas for improvement to increase the sustained use of ICS in the focus communities. This paper uses the methodological approach from the qualitative Technology Implementation Model for Energy (TIME) for the data collection and analysis elements. Our core results show a direct need for improved communication between all key stakeholder groups, the impact of demand and supply side financial incentives in creating reputational risk for community-based key stakeholders, and how the RBF mechanism promotes initial end-user adoption but not sustained use of ICS due to a focus on immediate results.

**Keywords:** results based financing; improved cookstove; modern energy services; sustainable development goals; energizing development; nepal

## 1. Introduction

Situated in the heart of the Himalayas, Nepal is a geographically, culturally, and societally complex country that has a long history of energy-based International Development assistance [1]. Despite an abundance of natural resources for producing sustainable clean energy through hydropower [2,3] and a robust Renewable Energy Subsidy Policy [4], 65.8% of rural households [5] use firewood to satisfy their cooking energy needs. When adjusted for income, this represents 67.2% of the poorest quintile and 19.8% [5] of the richest quintile. However, this national survey does not take into account the households who stack, or use multiple cooking technologies/fuel types concurrently, to satisfy their energy needs [6,7]. Robinson, et al. [8] detail this stacking phenomenon in the Nepal context in addition to outlining common barriers to adoption and sustained use of ICS across the value chain. The energy context in Nepal is set against the United Nations Sustainable Development Goals (SDGs), which champion a "Sustainable Future for All" [9]. SDG7 focuses on Energy with four core indicators, universal energy access, increasing the share of renewable energy, access to clean cooking and doubling the rate of energy efficiency. Unfortunately, whilst 94% of Nepal's population has access to electricity [10], the supply is often unstable and the infrastructure not suitable for households to rely on electricity for their cooking needs [11]. This results in only 29% of the population having access to clean cooking fuels and technologies [10].

Practical Action Nepal's (PAN's) Results Based Finance for Improved Cookstove Market Development (RBF) [12] project looked to develop the market for Improved Cook Stoves (ICS) in Province 3 and Gandaki Province of rural Nepal, situated 200 km west of Kathmandu, with the aim of deploying 40,000 Tier 2 and Tier 3. Results based finance is a payment mechanism that releases funding to the co-ordinating partners based upon results (at the end) rather than the traditional payment structures, where payments are released as costs are incurred and the work is completed [13,14]. This means that PAN have to fund the initial cost of the program with reimbursement after the program is completed. RBF1 focuses on Tier 2 ICS with the follow-on program, RBF2, focusing on Tier 3 ICS. ICS are categorised by performance using an internationally recognised testing methodology in 5 tiers, tier 0 being an open fire and tier 4 an electric hob, [15]. In Nepal there are 42 household and institutional approved ICS [16]. PAN looked to increase demand for ICS through offering increased customer choice by building the capacity of market chain actors, strengthening support services, and facilitating the enabling environment [17]. The main modality of PAN's methodology was Results Based Financing (RBF) [13,14], structured on a number of factors, including stove performance (tier level), warranty, and remoteness of the intervention area. Supply side incentives were provided to the private sector including suppliers of stoves and local financial institutes for last mile distribution. Demand side strengthening included finance made available for end-user discount incentives (for tier 2 and 3 ICS), behavioural change campaigns, and targeted assistance for private sector actors, municipalities, and local financial institutions.

This paper provides a qualitative evaluation of RBF using the Technology Implementation Model for Energy (TIME) developed by the authors, with a focus on highlighting key stakeholder voices. The aim of this paper is to better understand the key stakeholders' roles in creating the enabling environment for behavioural change around open fire cooking, resulting in end-users sustainably using ICS through a Results Based Finance (RBF) model. The four Research Objectives (RO) are to:

1. identify the barriers, enablers, and resulting engagement strategies for the adoption and sustained use of T2 and T3 ICS;
2. map out the role of key Stakeholders in the RBF Project using the TIME Methodology;
3. understand the relationships between key stakeholders and how they influence the enabling environment for behavioural change;
4. identify and rank areas for improvement with regards to increasing sustained use of ICS.

The novelty of this paper is twofold, first, the application of TIME to the Nepali context brings an alternate approach to an established international development sector. Second, TIME brings a key stakeholder driven approach to the evaluation of RBF. The limited literature surrounding RBF evaluations are singularly focussed on top-down methodologies where the end-user and community perspectives are not taken into account. This results in the propagation of programming that does not account for the perspectives of the intended beneficiaries of RBF. This research is also of central importance to documenting the complex contextual factors associated with the adoption and sustained use of ICS as well as creating an evidence base for future research and to ensure that any missteps documented throughout this paper are not repeated.

The structure of this paper is divided into three sections: Methods, Results and Discussion, and Conclusion and Recommendations. The Methods Section outlines the TIME key steps and decisions in the qualitative approach to data collection and analysis. The Results and Discussion Section bring attention to the core findings presented in accordance with the TIME methodology as well as highlighting a number of methodological limitations. Lastly, the Conclusion and Recommendations Section draws the paper to a close and presents a number of recommendations based on the data to PAN, which will increase the effectiveness of RBF.

## 2. Methods

The data collection in Nepal was conducted between January and April 2020, with the majority of transcription and analysis conducted in the following months. The following section outlines TIME as an evaluative tool as well as presenting the methodological steps to create a robust data and insightful set.

### 2.1. The Technology Implementation Model for Energy (TIME)

TIME provides a formulative and evaluative qualitative, stakeholder-orientated, multi-level analysis tool for the implementation of humanitarian energy technologies. The analysis framework is shown in Figure 1. TIME brings to light the complex social, environmental, and economic contextual issues that that can often destabilise humanitarian technology implementation as well as defining and identifying the impact of key stakeholder group actions on the behavioural change of end-users. These issues are contextually specific; TIME does not provide common issues but a methodology to discover and analyse the context. The model was derived from a number of existing models in the Water, Sanitation and Hygiene (WASH) [18,19], Social Enterprise [20,21], Appropriate Technology [20,22], and International Development literature [23,24]. A detailed background and definition of all terms can be found in Robinson [25]. TIME was then tested against a number of Global Challenge Research Fund [26] projects to shape theory from practice. Echoing the methodology of creating a heath orientated behavioural change model by Dreibelbis, et al. [18], TIME was modified based upon the results from this practical application.

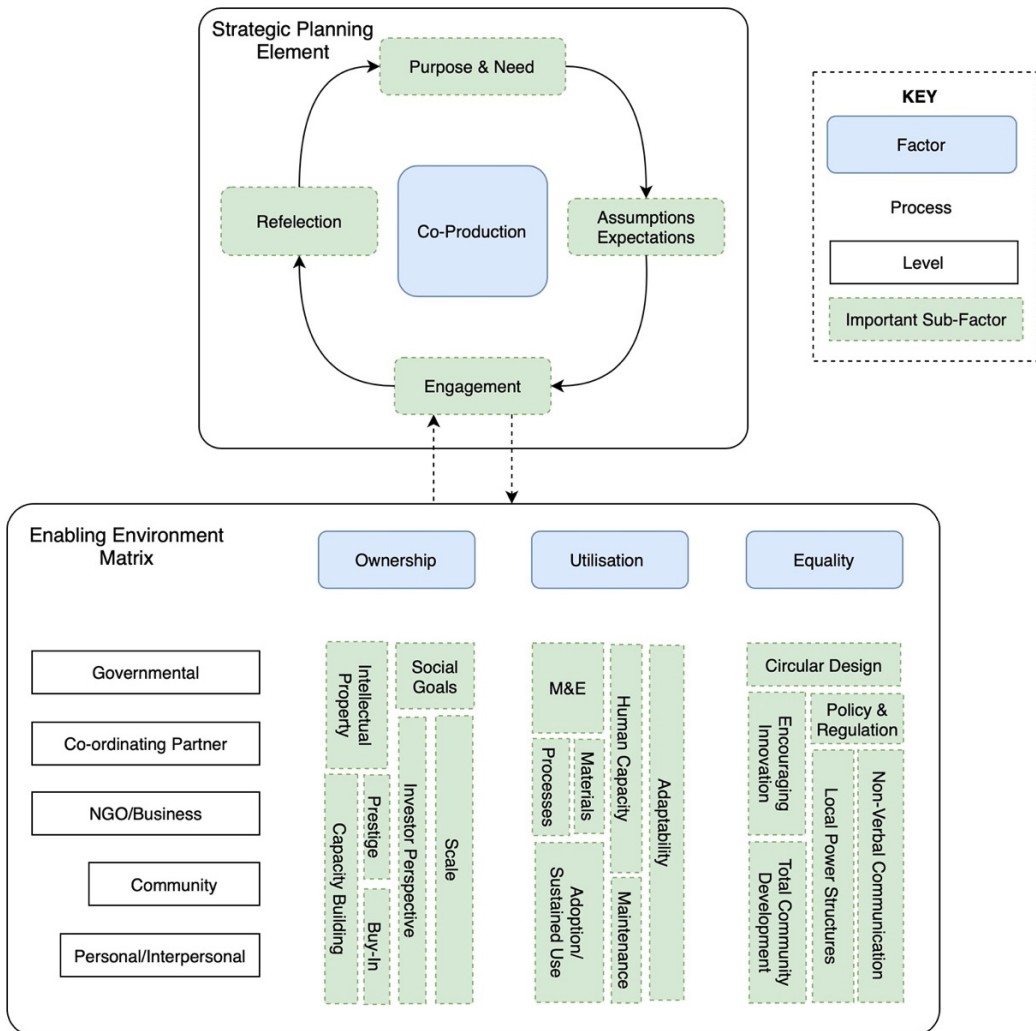

**Figure 1.** The TIME Framework.

TIME is divided into two elements, the Strategic Planning Element (SPE) and the Enabling Environment Matrix (EEM). The SPE aligns the project outcomes with the needs of the technology end-users and other Key Stakeholder Groups (KSG) through considering four co-produced sub-factors: Purpose and Need, Assumptions and Expectations, Engagement, and Reflection. The Purpose and Need factor identifies factors which influence behavioural change from the perspective of each KSG. These can range from willingness for users to pay for technologies to pre-existing cultural traditions. The Assumptions and Expectations sub-factor identifies the misalignment of key stakeholder assumptions and end-user expectations. The Engagement sub-factor identifies the programmatic engagement strategy, which links the causal pathway structure of the SPE and matrix structure EEM. The relationship between these two elements is cyclical as decisions in one will have impacts on the other. The final SPE sub-factor, Reflection, provides an opportunity for modifying the strategy based upon key stakeholder feedback.

The EEM defines the role of each KSG as well as visually mapping the interactions between these KSG. The KSG or levels are: Government, NGO/Business, Co-ordinating partner, Community and Personal/Interpersonal. These groups are deemed critical to the sustained use of poverty-alleviating energy technologies and are mapped across three core factors which influence the adoption and sustained use of poverty-alleviating energy technologies: Ownership, Utilisation (further divided into People and Systems and Physical Resources), and Equality. These factors are defined in Table 1.

**Table 1.** TIME Definitions [25].

| Factor | Definition |
| :---: | :--- |
| Co-production | The Key stakeholder groups co-designing and collaborating to produce outcomes that are of value to both the beneficiaries and KSGs |
| Ownership | Creating buy-in from the beneficiaries which goes beyond participation in the intervention |
| Utilisation of Resources | Utilising local people and processes to produce part of or the entire involved technology |
| Equality | An assurance that co-produced values or the perception of those values are equitable and just for all across the entire project cycle |

*2.2. Data Collection*

The TIME methodology is based upon a phenomenological approach to qualitative research where the "lived experience" [27] of the beneficiaries and the meaning behind why people make decisions is of key importance [27–29]. TIME utilises a qualitative data collection methodology divided into four sections: semi-structured interviews, focus groups, informal conversations, and semi-structured observations. The semi-structured interviews and focus groups used a semi-structured interview guide (the P1 and P2 semi-structured interview guides can be found in the Supplementary Materials) which was developed for the GCRF interviews and modified in line with the outputs of PAN's PARBF. Additional informal conversations framed the semi-structured interviews as well as reducing the effect of outsider status [30], as further discussed in Section 3.3.1. The observations were used to clarify user claims, for example if the end-user stated they used the ICS every day yet there was no soot blackening or firewood stacked close to the ICS then the interview information and observations did not support each other. Given the importance of accurately capturing the lived experience of all key stakeholders, study participants were selected based upon advice from Practical Action Nepal on who the KSG were both in Kathmandu and in the field as well as through the lead authors' previous experiences of field work in Nepal. All of the interview participants were closely involved with the RBF project in a range of roles, which represented a socio-cultural cross-section of ICS actors. These are summarised in Table 2. The interviews were divided into two phases: Phase 1 (P1) and Phase 2 (P2). P1 involved all Kathmandu based key stakeholders and P2 involved all field based key stakeholders. By separating the interviews by geographical location, P1 generated project

perceptions, whilst P2 identified the end-user realities highlighting any misalignment of understanding.

**Table 2.** Key Stakeholders.

| | |
|---|---|
| **Government** | 1 × National Government (AEPC), 2 × Local Government (Myagdi and Baglung) |
| **NGO/Business** | 1 × Local NGO, 3 × Improved Cookstove Manufacturers, 1 × Micro Finance Organisation |
| **Co-Ordinating Partner** | 1 × Co-ordinating NGO (Practical Action × 2) |
| **Community** | 1 × Heath Worker, 1 × Community Forestry Representative, 1 × Local Financial Cooperative, 3 × Local Distributors |
| **Personal/ Interpersonal** | 4 × Tier 3 ICS Users, 3 × Tier 2 Users, 4 × Non-ICS Users, 2 × User Focus Groups (Tier 2, Improved Traditional Cookstove and Traditional Cookstove Users), 4 × Informal Interviews with T2 Users |

The P1 interview guide was divided into four sections reflecting the structure of TIME made up of "open" questions as defined by a phenomenological qualitative approach [31] allowing the interviewee to shape the nature of the results. The first gathered contextual data such as background information, role, gender, age, and details on the organisation they represent, if applicable. The second, explored the four strategic planning elements (Purpose, Assumptions/Expectations, Participation/Engagement, Reflection) through the lens of co-production. The third focused on the KSGs included in the EEM, looking to understand KSG roles and interactions. Finally, emphasis was placed on understanding how the KSG integrate the three factors of behavioural change (ownership, utilisation, equality) across the five core levels. Given the complexities of conducting field visits in Nepal, due to the remoteness of working communities and the need to inform the relevant field-based stakeholders, the P1 interviews were completed, transcribed and analysis started before the P2 interviews were conducted. It was also important for PAN to combine this data collection visit with other work to reduce the cost of a field visit. This resulted in the semi-structured interview guide for P2 (the community, end-user, local government, and local NGO interviews) being shaped by the initial results from P1 which provided information on the perceived barriers to cookstove intervention and the biggest behavioural change challenges. The P2 interviews thus provided user/community perspectives on these barriers, either capturing a different set of barriers, reinforcing the same barriers, or a combination of the two. All interviews (P1 and P2) finished with an opportunity for the participant to ask any questions or talk about any relevant areas that they felt were overlooked.

This study was approved by the Ethics committee at the University of Nottingham. In order to comply with these Ethical Research Guidelines, all participants were shown and asked to read the pre-interview information sheet and asked any questions they had before the interview was conducted to ensure that they were comfortable with the process, and that they could withdraw any point without penalty. All interviewees signed a consent form that allowed their data to be used as part of this study.

*2.3. Data Analysis*

The analysis of data was divided into a number of parts in accordance with TIME. First, we coded SPE data in line with the four sub-factors (Purpose and Need, Assumption and Expectation, Engagement, and Reflection) effectively mapping the Barriers and Enablers (B/E) and Engagement Strategies (ES) to overcome the determinants in line with RO1. In addition, the reflection element supports RO4 as well as the observations made in the field visits in March 2020. Second, we coded data into the EEM through the three factors (ownership, utilisation, and equality) and five levels (Government, NGO/Business, Co-Ordinating Partner, Community, Personal/Interpersonal) showing the perceived roles of each KS from the perspective of individual KS in line with RO2 and 3. All coding was

conducted by the lead author using Nvivo12 [32], in accordance with the University of Nottingham ethical approval to protect interviewee data (see Section 3.3). As the volume of data was large, this allowed easy classification and identification of nodes and cases for the first stage of coding. No pre-existing coding framework was used as it was important that emerging themes were driven by the interviewees not the interviewer. This reflected the nature of the open-ended questions asked through an inductive approach [28,29].

For the SPE P1 and P2 were treated as separate collections of data, which meant that we did not apply the coding framework established in P1 onto P2 but started the inductive process from the beginning. Again, this was to highlight any differences between P1 and P2 in the phrasing or language used by the two groups of interviewees. After the nodal frameworks were established the second stage of coding was to run through the nodes and confirm that firstly, they were correct and secondly, the definition of each node was correct, whilst removing any repeated nodes to increase the robustness of the results. Following this, the B/E and ES identified in P1 and P2 were compiled into a matrix which ranked the B/E and ES on number of KSG mentions—a rough importance guide. The Reflection and Assumptions and Expectation elements were also coded using an inductive method to build a case for RO4.

Framed by the EEM, the second part of the analysis, understanding the role of KSG, captured the unique perspectives of the KSG in the PAN RBF project. This determined what each stakeholder believes their role to be and how they interact with other KSG. The remaining three factors—ownership, utilisation, and equality—were used as the framework for coding. The data was coded into both levels and factors, for example if a KSG mentioned the role of government policy influencing local manufactures it was coded (Government, Utilisation (people and systems)). This coding system produced a matrix for each KSG, the nature of this data distribution provided an indication of how each KSG perceived the project when coupled with supporting quotes.

## 3. Results and Discussion

The following section outlines the results of the 31 semi-structured interviews from the P1 and P2 data collection and analysis. In line with the TIME elements, first, we consider the SPE results and implications of these on the RBF project. Second, we discuss the EEM results and discussions around the impact of these findings. Due to the volume of data produced by TIME, in this paper we have extracted key learnings and themes in line with the research objects co-produced with PAN.

### 3.1. Strategic Planning Element

The presentation of results echoes the four sub-factors contained within the SPE in Figure 1. In this section we discuss how the purpose aligns with the beneficiary needs through the identification of B/E across the two data collection phases. We also identify what assumptions KS made and the impacts of these on user expectations. We then consider the engagement strategy when interacting with KSG, paying careful attention to the impact of this engagement strategy on end-user behavioural change. Additionally, it is important to reiterate that the results of P1 show the perceived B/Es from the perspective of Kathmandu based KSGs, whereas P2 shows behavioural determinants as identified by end-users.

#### 3.1.1. Purpose and Need

Contained within the Purpose and Need sub-factor, 65 behavioural determinants emerged from the coding and analysis of 1486 data points; the top 10 from each phase are summarised in Table 3. In contrast to Dreibelbis, et al. [18], who grouped B/Es into contextual, psychosocial, and technological factors, we have ranked the 65 B/Es by KSG mention. In further developments of TIME this may be helpful for exploring the contextual issues in more detail.

**Table 3.** Top 10 Barriers and Enablers.

| Ranking | Phase 1 (Perceptions of B/E) | Phase 2 (Actual B/E) |
|---|---|---|
| 1 | Awareness\Do not understand benefits | Convenience and Stacking |
| 2 | Finance\Willingness to Pay | CS Use\Heating |
| 3 | CS Use\User Experience | Finance\Cannot afford ICS |
| 4 | Convenience and Stacking | Aspiration |
| 5 | Historical Use—living in traditional way | CS Use\Smoke and Health\Smoke affecting health |
| 6 | Aspiration | CS Use\Time Saving\Time saved cooking |
| 7 | Time Saving\Time (not) saved preparing fuel | Availability of other Tech. |
| 8 | CS Use\User Friendliness of Tech | CS Use\Firewood or Biomass Fuel\No shortage of firewood (collection from own land) |
| 9 | Social Status | Awareness\Understand benefits of ICS |
| 10 | Finance\Other financial priorities | CS Use\Taste of food better with wood |
| 10 | Dependency | CS Use\Firewood or Biomass Fuel\ICS uses less firewood |
| 10 | No Supply Chain\Pellets | - |

The first difference in perceptions between P1 and P2 was that the P1 KSG have a different perspective of what is important to the end-user. Ranking 1st in P1 as the biggest barrier to adoption is that users do not understand the benefits of cooking with an improved cookstove:

*"the awareness among the user is still not adequate. They are not understanding why this cook stove should be in their kitchen. That awareness still has not been created enough. Unless the user understands it, it is doomed to fail" (NGO/Business).*

However, it became clear that all 17 of our P2 Personal/Interpersonal KSG clearly understood the benefits of using an improved cookstove; this group included both ICS end-users and non-users. Interviews with them indicated that the gap was not in awareness, but around basic training given to the ICS end-users:

*"Many people have not used it because they did not know how to use it. There should be some monitoring teams who should come over, and if they see such situations, they should teach us how to properly utilize it. But nothing like this happened. They just did it for sales" (Personal/Interpersonal).*

This mismatch in key stakeholder understanding is often quoted as a repeating failure in the ICS literature [33]. The second core finding from P2 centred on the convenience of each cooking technology resulting in end-users stacking technologies [7,8,34]. Each interview in P2 stated that no cooking technology satisfied all their needs thus people used multiple technologies whilst cooking one meal, illustrated in the following quote:

*"We cook in an improved cook-stove [mud brick]. After that, daal is made on gas [LPG] in the pressure cooker. And then I cook the vegetables outside in improved metallic cook-stove. After that in winter, water is boiled in "Bhushe" cook-stove [sawdust Tier 2] and we bathe from it. When we cook for the goat we use the "Taulo" [Three Stone Fire]. If we have to cook flat-bread, I think now I should use this [Tier 3 Metallic Cookstove] to make dry flatbread" (Personal/Interpersonal).*

Figure 2 shows a typical kitchen from P2. This interviewee stated that there were seven different cooking technologies being used with three energy sources (LPG, kerosine, wood): LPG hob, an open fire, improved mud and brick stove, two tier 2 metallic cookstove, tier 3 metallic cookstove, and a kerosene stove. The motivation for the use of each of these technologies emerges through the Purpose and Need sub-factor; the three stone fire (TSF) and improved mud and brick cookstove provide heat during the winter as well as a larger

scale cooking option, such as cooking for cows and goats. LPG stoves are quick, they are used for making tea and entertaining guests (to convey social prestige as well as reducing smoke in the home) but tend to be used sparingly as gas bottles cost at least 1500 npr (15 USD) to refill. LPG is also used when other fuel sources/improved technologies are not available as a last resort. Whilst this phenomenon of stacking is not exclusive to the Nepali context, it seems to be exacerbated in Nepal due to cultural cooking traditions, International Development assistance, and a robust biomass energy policy [4]. The time saved whilst cooking on any technology was viewed as important for aspirational reasons linked to social prestige and a desire to free up time for leisure, which reflects the role of social media in helping to promote an increased desire for improving the quality of life:

> *"People now-a-days seek luxury. Not only people from cities but people from villages also yearn for luxury. Maybe it is also due to foreign employment. Now, in the villages all the agricultural lands are on the verge of being unproductive, as people do not want to work in the fields. Everybody uses gas, electricity is being used for rice cooker and even to boil water. So people are yearning for pleasure, that is the reason [ . . . ] It is because people seek luxury. People want pleasure"* (Personal/Interpersonal).

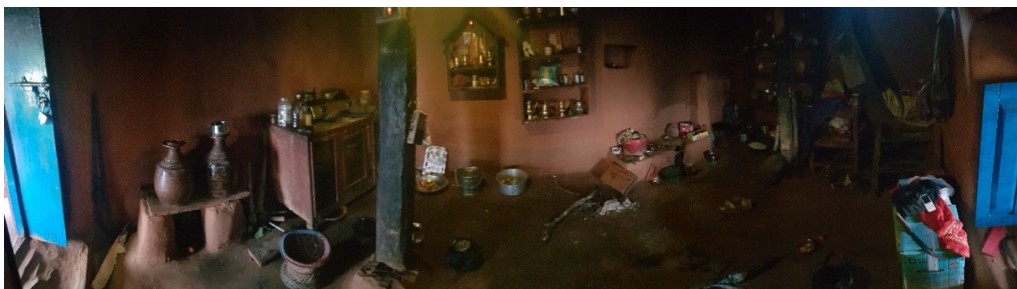

**Figure 2.** Typical Nepali Kitchen.

P2 also showed that users understood the impact of smoke on health, "we can get a more advanced and better stove than this which will not emit smoke, which will also protect us from diseases" (Community), however these long terms risks were "backgrounded" [7] by priorities, such as a need for space heating, preferences for the taste of food cooked on a TSF or improved mud ICS biomass, or a desire to utilise an abundance of firewood rather than paying for LPG. Every community in P2 was part of a community forestry group responsible for the sustainable management of local forests.

Lastly, finance appears at 2nd and 10th place in P1 and 3rd in P2. However, the "willingness to pay" seen in P1 does not correlate with the "can't afford ICS" in P2. P2 KSG state that whilst the minority of community members do not have the capacity to pay for a cookstove, the majority are unwilling to redirect the small amount of income they earn to an ICS. A P1 key stakeholder explains this process, *"they [end-users] think it is absurd to buy a stove for Rs 2-3000 when you can make it using some stones, bricks and mud for 100 or 200 rupees [ . . . ] They are not health conscious but financially conscious" (NGO/Business).* One P2 stakeholder also reinforced this, *"It is not because people cannot spend money, there could be some like 2–4 people out of 100 who cannot afford it" (Community).* However, all P2 interviewees were nevertheless concerned about the price of the cookstove. The misunderstanding of end-users' financial priorities is also documented by Rhodes, et al. [35] and Hulland, et al. [36] in the ICS and WASH sectors, respectively.

### 3.1.2. Assumptions and Expectations

22 different Assumptions and Expectations emerged from the analysis, of which the top 10 by mention can be found in the Supplementary Materials. The underlying assumption of RBF1 and 2 was that users want cookstoves but cannot afford them; a common assumption in ICS programs driven by International Development actors [37]. Whilst this does appear from the P1 interviews, results from P2 would suggest that the situation is more complex than it seems and that the majority of potential users either

have cooking solutions that already satisfy their energy needs and/or cooking is not an investment priority for households. The reflection sub-factor illustrates how these assumptions and expectations have impacted the project.

### 3.1.3. Engagement

The Engagement sub-factor takes into account the previous two sub-factors and builds an Engagement Strategy (ES) that adequately satisfies the technology end-users' needs, the co-ordinating partners purpose and also takes into account the assumptions and/or expectations. Table 4 outlines the top 10 (out of 33 ES and 716 data points) by mention. Unsurprisingly, in a program about supply chain strengthening the top ranked RS in P1 was "Supply Chain Strengthening". This builds upon the assumption that "Users want ICS" (ranked 2nd in P1 assumptions) and that there is not sufficient capacity on the supply side to meet the demand. However, this is contradicted by the P1 top ranked behavioural determinant that users do not understand the benefits of ICS—this would suggest that there is a low demand due to a lack of understanding on the demand side, not a lack of supply. PAN recognise this contradiction resulting in the local NGO, distributors, and community groups completing awareness campaigns (Ranked 1st in P2) through a number of mechanisms that appear important in both P1 and P2. These include the use of "Formal or Informal Peer to Peer marketing", "social media marketing", promoting local products and fuels, and leveraging the impact of the Indian 2016 LPG blockade (information on Indian Fuel Blockade—https://www.bbc.co.uk/news/world-asia-35041366 (accessed on 11 May 2020). The focus on both demand and supply side elements is a trait common to market orientated poverty alleviating technology initiatives [38–40].

**Table 4.** Top 10 Engagement Strategies.

| Ranking | Phase 1 | Phase 2 |
|:---:|:---:|:---:|
| 1 | Supply Chain Strengthening | Awareness Campaign\Communicating ICS Benefits |
| 2 | Awareness Campaign\Communicating ICS Benefits | Mobilize Financial Institutions |
| 3 | (Government) Policy and Subsidy\Incentive Scheme (Coupon System) | Formal or Informal P2P Marketing\Recommendation from friend or Community leader |
| 4 | Awareness Campaign\Cookstove Demonstration | Awareness Campaign\Cookstove Demonstration |
| 5 | (Government) Policy and Subsidy\Reduction in ICS Cost | (Government) Policy and Subsidy\Reduction in ICS Cost |
| 6 | Modifications of Tech. to Satisfy User Need | Formal or Informal P2P Marketing\Volunteer Distributor |
| 7 | Formal or Informal P2P Marketing\Recommendation from friend or Community leader | (Government) Policy and Subsidy\Providing documents |
| 8 | Mobilize Financial Institutions | Blockade |
| 9 | Habituate Technology | Social Media Marketing |
| 10 | Warranty and Maintenance | User buying from Local Market |
| 10 | (Government) Policy and Subsidy\Local Manufacture Preference | Formal or Informal P2P Marketing\Through community groups |

In terms of responding to the financial barriers and enablers there are a number of strategies that are being used across a number of societal levels. Independent of PAN's RBF program. Firstly, the National Government led Renewable Energy Subsidy Program which [4] which results in the reduction of the ICS price at the consumer level [8]. The second strategy is the mobilisation of financial institutions through incentives provided by PAN (which reflects the structure of the Government of Nepal subsidy program) and using

the social impact of this program as an additional persuasive tool; however, the reflection sub-factor showed there was a reputational risk of financially incentivising local financial institutions, which will be explored further in Section 3.1.4. This is an impact of the RBF method that has not previously been identified in the literature due to previous evaluations having limited integration of end-user and community voice.

The final engagement strategy considered in P1, but not P2, was the habituation of technology or its integration into the user's daily routine was not considered by the users as, if it is convenient, it will be used and if it is not, it will not be used. Technology habituation is captured through TIME by the inclusion of elements from the Domestication Framework [41] and Behavioural Settings Theory [42]; both of which are built upon in WASH and more general health theories as outlined in Robinson [25]. This is deemed an essential element for the sustained use of poverty alleviating WASH technologies and due to the interdisciplinary nature of poverty alleviation [43] is critical to ICS programming.

### 3.1.4. Reflection

The final sub-factor in the SPE is Reflection, presented in Table 5. These reflections are important to identify areas of improvement as well as giving all KSGs the power to influence and co-produce current and future programs. Not all reflections were based upon areas of improvement; many were complementing the positive aspects of PAN's RBF and how RBF2 had built upon the successes RBF1, such as:

> *"There is a big difference because previously the diseases inflicted by smoke like COPD, Lung diseases, pneumonia in kids have dramatically declined after using the modern cook stoves"* (Community).

> *"The cook stove that has been distributed from this organization has given us a sigh of relief because people are not littering ashes here and there and the consumption of wood has gone down, and it is also a bit beneficial for the environment and for health"* (Personal/Interpersonal).

> *"What I like about this project is that you are not promoting certain type of stoves. Actually, you are giving choices to the user. And based on their willingness, the model they would like they are buying the stoves [ . . . ] Users getting choices to choose the program is the unique thing about this project"* (NGO/Business).

First, in terms of supply chain strengthening, there is no supply chain for the fuel for the tier 3 stove (pellets) which results in them being incorrectly fuelled. This has led to poor performance and discarding of the technology in the RBF2 communities. Further, the ICS users, being receivers of technology, do not reflect on the project goals or systems, just the technology itself. These reflections are an extension of the barriers and enablers. For example, one of the barriers is convenience with some users reflecting that the T3 cookstove was not convenient enough.

Second, the incentive to local financial institutions was given per cookstove that they were able to sell. The opinion of a number of P2 KSGs was that the institution forced its members to purchase the cookstove, *"if they are the member of it [the financial institution], it is mandatory for them to get it [the cookstove]"* (NGO/Business), which resulted in a lack of support and training from the users, *"They just said that okay, the cook-stove has arrived, if you want to take it, come. The one who has the money, will take it, that's it. They didn't even talk about its benefits and negative effects"* (Personal/Interpersonal). This highlights an underlying communication issue which emerged in a number of areas. Additionally, end-users were incentivised through a voucher system. If the end-user attended a cookstove demonstration and was interested in purchasing the cookstove then a voucher would be given. However, this process worked differently in reality as attendance at the demonstration was not mandatory:

> *"we also have a token, remember the one we showed you yesterday [ . . . ] If you are interested you can take the token and buy the cook stove [ . . . ] We have said if you do not have money to buy the cook stove we will provide it"* (Community).

**Table 5.** Top 10 Reflections.

| Ranking | Phase 1 | Phase 2 |
|---|---|---|
| 1 | Problems with subsidy system or incentive | User has no communication with local NGO (M and E) |
| 2 | Improvements, Feedback for ICS | Improvements, Feedback for ICS |
| 3 | RBF1 to RBF2 improvements | User not knowing how to claim warranty |
| 4 | There is duplication of programs | User Perspective\Feel cheated by distributor (financial co-operative, etc.) |
| 5 | Positive Impacts of RBF | User has no communication with local government |
| 6 | People with money buy, people without money do not | Positive Impacts of RBF |
| 7 | User has no communication with local NGO (M and E) | User not taught to use or build ICS effectively |
| 8 | User has no communication with local government | Problems with subsidy system or incentive |
| 9 | Focus on adoption rather than sustained use of ICS | Communication of Funding Systems to Users |
| 10 | Government does not understand ICS programs | User does not know anything about ICS program |
| 10 | Manufacturer implemented suggested changes | - |
| 10 | Other KS involved in improving ICS | - |
| 10 | Manufacturers not involved in M and E | - |

Next, it is difficult for the users to understand the system that reduces the cost of the cookstove. At the point of sale, the users are presented with a price, not an explanation of how that price was achieved. When projects are duplicated through different organisations in the same geographical area, end-users may see the same technology for significantly different prices. This influences their choice to purchase new technologies as many projects give away technology for free. This issue of duplication should not happen as all energy projects are meant to be approved by the Alternative Energy Promotion Centre (AEPC, a government body) [44] and the local government, so the conclusion to draw is that this process does not stop duplication. The duplication of energy projects has another significant side-effect in relation to the distorted perception of value from the end-user perspective. In RBF2 the tier 3 cookstove costs 9000 Npr (according to manufactures), however it is being sold to the users at 2500 npr. In RBF 1 the tier 2 cookstove was priced at 3000 npr but was found in the local market by a number of users for 1200 npr. This led users to ask, *"when cost is 1200 nr in the market why are they taking 3000 npr?"* (Personal/Interpersonal) and resulted in users not adopting or using ICS, a core goal in the RBF program. The other P2 KSs feel the results of this as, due to the results-based nature of the funding mechanism, there is a pressure for immediate results, through the purchase of ICS rather than sustained use. One P2 KS summarised:

> *"Yes, if the results are not visible right now, it does not mean it is not a success [ . . . ] But we want immediate results like we are given a target of distributing 'x' amount of stoves in 2 years' time [ . . . ] We want quick fixes. We are asked to meet our target and distribute 'x' amounts of stove and get the money to pay our staff and management"* (NGO/Business).

This also affects the quality of monitoring and evaluation as there are no resources allocated to this.

> *"When it comes to the areas in RBF 1, there was no monitoring because previously we had the program so we went there, but now we are not related with the program. But if the RBF 2 program will be conducted in our past working areas, the monitoring will be done automatically"* (NGO/Business).

*"We do not have the financial prowess to organize programs but what we are doing is, we reach out to people when they gather, for instance, at co-operative meetings, fairs etc and try to spread the information about the benefits cook stove" (NGO/Business).*

Our last communication reflection shows that not only does this short-term view impact the local NGO, distributors, and community groups, but it dictates the feedback mechanisms from the P2 KSG perspective. There is no time or system for the end-users to communicate with the levels above them (ranked 1st, 5th, 7th, and 9th in P2 and 7th, 8th, and 10th in P1). This also has an effect on "users not knowing how to claim the warranty" (3rd) or "adequate training around using the cookstove" (7th). This is reflected in the literature through the comparison of top-down vs. bottom-up approaches to development initiatives. Haney, et al. [45] outline this issue for the electricity sector, stating "the recent top-down approach of electricity provision largely informed by a "meeting-basic-needs" paradigm is vulnerable to applying one-size-fits-all solutions to communities with different and often more sophisticated energy demands than what may meet an outsider's eye" (p.7).

The final reflection is not contained within Table 5, but still remains important—*"What people want is the organization should provide it for free, and people are willing to use it if they get it for free" (Personal/Interpersonal).* This contradicts the core values of the market element of the RBF as users seem not to be using the cookstoves when they pay for them. It is not logical to assume that use will increase if they are provided for free.

### 3.2. Enabling Environment Matrix

The unique EEM establishes key stakeholder role perceptions and key stakeholders' interactions. Whilst the EEM enables the practitioner to dive deeply into the perceptions of each key stakeholder group by creating an EEM for each KSG, in this section a number of overall trends and themes will be outlined from the 392 data points. For specific role perceptions of KS groups refer to the Supplementary Materials.

Figure 3 shows how the core factors are distributed amongst the KSG from all KSG perspectives where the NGO, Business KSG is perceived to have the most important role in RBF. This is no surprise given the supply chain strengthening aspects of this project. What is surprising is the lack of a perceived role for the co-ordinating partner as there were zero mentions from the personal/interpersonal level about the co-ordinating partner and only a few from the other KS groups. Given that the co-ordinating partner manages all the KS groups, there was a distinct lack of visibility.

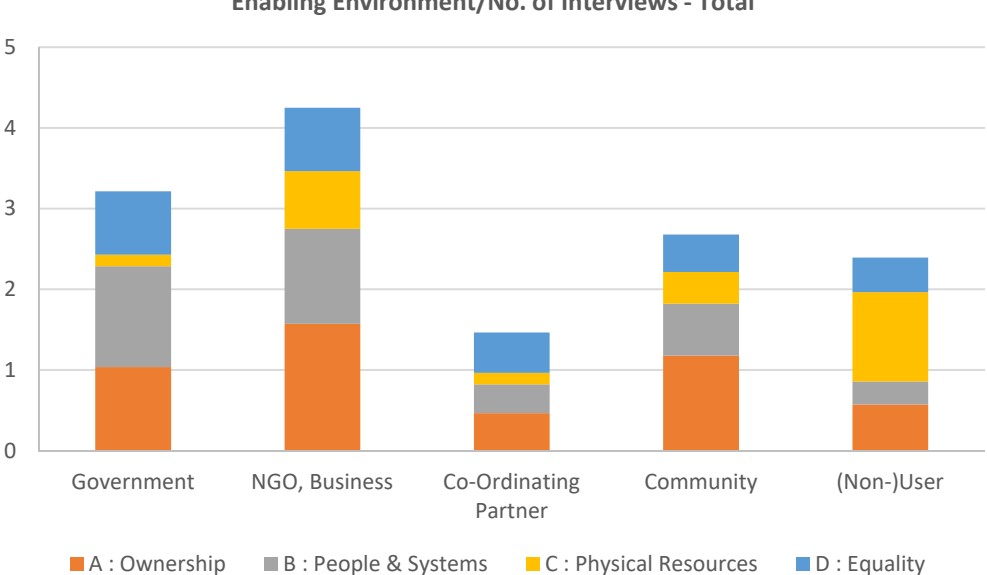

**Figure 3.** Total Factor/Level Breakdown.

Whilst Figure 3 shows the distribution of data across the key stakeholder groups and core factors, highlighting the importance by mention of key stakeholders and core factors, Table 6 shows the specific perceived roles of each KSG from, in this case, the perspective of the user and non-user KSG. By mapping the roles of each KSG from each KSG's perspective the perceived role of each stakeholder is established and the misalignment between expectation and reality is identified by comparing multiple perspectives. For example, as mentioned above, there is no perceived role for the co-ordinating partner from the perspective of the personal/interpersonal KSG prompting a change in communication method. In addition, roles ordinarily carried out by the co-ordinating partner, such as awareness campaigns and assessing needs, are completed by the community from the perspective of the other KSG which also required a modification of communication strategy.

**Table 6.** Personal/Interpersonal EEM Perspective.

| | Ownership | Utilisation | | Equality |
| --- | --- | --- | --- | --- |
| | | **Human and Systems** | **Physical Resources** | |
| **Govt.** | Local Govt. Programs (energy, farming, infrastructure, etc.) | Assessing Need (or not) | | |
| **NGO/Business** | Cookstove Promotion Social Media Marketing Subsidy Dissemination | Communication with User (Or Not) M and E | Warranty | Preferential treatment to friends not needy Success of other projects |
| **Co-ordinating Partner** | | | | |
| **Community** | ICS Distribution/Awareness | Assessing Needs before starting project | Warranty through local distributor Community Forestry Group | Co-Operative Loans Reputational Risk due to lack of communication |
| **User** | Quality of Product and Service Recommended from Friend Providing Citizenship Card Seeking Luxury Investment in ICS | Lack of Communication on Subsidy System Willingness to pay Reliance on others for Technology | Technology Stacking Who will repair if it breaks? Firewood Collection Building ICS Themselves Dependency on import of LPG | Confusion over dissemination Migration Decreasing Birth Rate |
| **OTHER** | | | | |

Another trend is communication between KSG. This includes information transfer between KSG, as information is often disseminated by the co-ordinating partner in a top-down model with limited opportunity for feedback through bottom-up systems. For example, all of the personal/interpersonal interviews indicated a lack of opportunity to give feedback to either the local NGO or local distributors. The root of this issue is a confusion over responsibilities resulting in an "economy of no-knowledge"—a passing down of responsibilities to the KS that interact with the users, whether that is the community groups, local NGO, or local distributors. The first effect of this is that due to these undefined roles all other stakeholders think that the others should be doing more to help. The second effect is the reputational risk associated with disseminating cookstoves. There were three stakeholders who were concerned about this, due to the inconsistent pricing of cookstoves, communication regarding funding systems that reduce prices and support systems post payment. Groups that indicated such concerns were the users recommending the ICS to their peers, the local distributors, and the financial co-operative.

All KSG groups stated that the government should take a more active role in understanding the energy needs of the rural populations. Again, all P2 KSGs did not have an opportunity to talk to local government about their energy needs. One user stated, *"We are people from the educational sector, when they [government] do not have time to ask about the*

*school, there is no chance of asking about cooking" (Community).* Yet, when interviewing local government officials, the response was the same, *"they [the co-ordinating NGO] can bring different programs not only this kind" (Government)* with the responsibility on the co-ordinating partner to help the community, shifting the responsibility away from government. However, the government officials did offer to provide lists of marginalised people if they were approached, which they have not been.

The final trend was around monitoring and evaluation and where the responsibility of the KS ends in terms of cookstove dissemination. Monitoring is conducted by the local NGO (through the local distributors due to budget constrictions) to check if the ICS have been received but not to check if they are being used. This raised the question of who is responsible for ensuring that the ICS are being used effectively after RBF has finished.

*3.3. Research Limitations*

In addition to the methodological limitation of TIME stated by Robinson [25], a number of contextually specific limitations surfaced during this evaluation. First, it must be recognised that the 31 semi-structured interviews may not represent the views of all voices included in RBF as over 40,000 ICS were incentivised across RBF1 and 2. However, the local NGO was responsible for 22,221 cookstoves in the area where we conducted the interviews. We have tried to mitigate this limitation by asking the co-ordinating partner, Practical Action, to place the interview team in communities that give a representative cross-section of the entire project. Second, it is difficult to mitigate against the impact that other International Development projects have had on the communities in terms of successful or failed initiatives, which may result in differential treatment of interviewers. Next, the difference between perception and reality among the interviewees must also be recognised. The interviews conducted with KS shows the perception of the KS, however these perceptions may be influenced by pre-existing biases. Lastly, a number of the KS fit into 2 KS groups (for example, government official and user) so we mitigated this by defining clearly at the beginning of the interview which role we would like them to have during the interview.

3.3.1. The Role of Interviewer Bias, Positionality, and Outsider Status

Given the qualitative nature of this paper we acknowledge the influence of bias, interviewer positionality, and outsider status [30,46] on the results. The issues arising around outsider status were more prevalent in the rural setting due to the larger perceived disparity between socio-economic status. We tried to mitigate the impact of being outsiders by staying in local accommodation in the community and building trust over a longer period of time. In the Nepali context it is unusual for development practitioners to stay in the village that is the focus of the project as normally day trips are conducted from district headquarters. Also, being accompanied by a Nepali research assistant, even though he was from a different district, allowed conversation to occur in both formal and informal settings. The informal conversations, which occurred whilst eating and socialising with community members, resulted in deconstructing some Nepali preconceptions of Europeans and the lead author's own preconceptions of Nepali people and culture.

We also recognise two other key issues during the data collection, for a number of interviews conducted in Baglung a representative of the local NGO partner was present. Whilst it is difficult to measure the effect this presence had on the answers of the interviewees, this effect of this must be acknowledged; especially as occasionally the local NGO representative would finish the answer to a question in an honest attempt to fill in information rather that direct the interview. The second issue was highlighted when one during an interview, a member of a financial co-operative told a user what to say and did not allow the user to give negative feedback. In this case the co-operative member was asked to leave and again, we stressed the importance of open, honest feedback to the interviewee.

During the data transcription and analysis, it was important to involve the research assistant who was independent of Practical Action. This research assistant was responsible for translation during interviews and translating/transcribing interview transcripts from the recordings. Unfortunately, due to the complexities of translating Nepali, a second research assistant was needed to meet the project deadlines and a research assistant was supplied by PAN who had previous qualitative research experience. A total of 56% of transcripts were completed by the first research assistant and 28% by the second, while the lead author completed the remaining interviews conducted in English. In order to check transcription quality, both research assistants were asked to complete a number of the same transcriptions. This process of including three people, two of whom were present during the interviews, helped to mitigate positionality issues during the transcription of data. However, the lead author alone conducted the data analysis in accordance with the Ethical approval to protect interviewee data, so an element of positionality must be acknowledged. This had the additional advantage of ensuring that the interpretation of the data was consistent across the coding and analysis process.

## 4. Conclusions and Recommendations

This section brings together the results of the TIME analysis through a series of recommendations (RO4) for PAN, informed by the data collection and analysis as well as the lead author's field experience. These recommendations are divided into five groups: communication, the impact of incentives, understanding why end-users purchase ICS, the reusability of market chains, and the impact of focusing on adoption rather than sustained use. Completing the interviews in two phases also provided a unique opportunity to not only map the barriers, enablers, and best strategies to overcome these from the perspective of the community/users, but also from the perspective of the KS based in Kathmandu leading to a divide between perceived and end-user stated barriers and enablers. Reflecting traditional centralised Nepali power structures, the top levels of the EEM (government, NGO/business, and co-ordinating partner) are situated in Kathmandu valley, which is geographically, topographically, culturally, and contextually different to the rural bottom levels (community and user).

First, more effective communication methods are needed for both bottom-up and top-down information sharing to define who takes responsibility for each role as well as what is assigned to the role as shown by the SPE and EEM results. The lack of communication between KSGs was highlighted by the personal/interpersonal level of PAN's RBF program not having heard of PAN or the role they fulfil in the project, no understanding of the incentive system, and how it affected the cost of the cookstove resulting in a reputational risk for the local suppliers. In addition, the researchers were the first representatives from the program to be in contact with the participants resulting in a perception of no support. The lack of communication around subsidy also resulted in the tier 2 ICS being priced at 3000 npr and the tier 3 at 2500 npr (when the manufacturer's quoted price was 9000 npr)—a technologically superior product for less. This line of investigation prompted the question, does an incentive have a positive impact? And positive from whose perspective? From the perspective of the end-users the incentives drive down the cost of ICS, possibly increasing the likelihood of purchase. However, given the high number of international organisations promoting ICS in the same villages/districts, many potential users will just wait until they are given the cookstove for free. The incentives also distort the cost of technologies, resulting in a distorted value for money proposition where users expect more than is possible to supply. This results in the second observation, by associating with one program and not effectively communicating the incentive, local distributors, or financial co-operatives are seen as trying to profit off the end users; discrediting other local financial schemes outside of RBF in the process. However, despite many of these drawbacks the RBF methodology carries weight in international development funding.

Third, a better understanding of why people purchase ICS and what end-users value is needed. This is highlighted by the differing results from P1 and P2 of the SPE. P1 KS stated

simply that people do not understand the value and users need to be more aware. Yet, P2 showed definitively people understand value as there have been cookstove programs here for 15+ years. Community members want better service and support as it is not about a lack of finance for the majority but how conveniently the ICS fits into their existing cooking stacks. As also identified by Robinson, et al. [8], supply side subsidies structure around the Government of Nepal Renewable Energy Subsidy [4] system do not account for stacking due to their "one stove per kitchen rule" and the lack of multi-dimensional data capture for fuel use, which is a common theme through much of the ICS adoption and sustained use literature [6,7,34]. However, unlike previous studies many P2 users also stated that buying an ICS does not mean they will use it.

Next, RBF1 focuses on tier 2 ICS, RBF2 on tier 3 with plans to expand to electric induction hobs. All of these project phases have been directed at the same or geographically close communities prompting the question of whether the market chain can be used multiple times? Given the "one stove per kitchen rule" many community members asked why they should not just wait as there will be another better ICS available soon and they might as well use the subsidy to invest in the best ICS they could afford (T3) rather than buying a less good ICS and then wishing they could upgrade when a better one comes out. These questions need careful consideration and further consultation with community members to establish the correct action as there is currently no literature to reference these questions.

Lastly, the behavioural change elements of RBF2 suggest a transition from producing impact to changing behaviours, however, in reality only 5% of households were monitored for use which was not enough to establish sustained use. In one case, a local cooperative believed over 80% of tier 2 ICS were still in use, but our observations directly contradicted this. More emphasis is needed on supporting the sustained use of ICS by end-users, rather than the limited support given currently in the form of, at best, a cooking demonstration. In many cases this seemed like a tokenistic approach to behaviour change. In contrast, the WASH sector has transitioned away from technology (or hardware) to behavioural (or software) based approaches which look to better understand the drivers of behavioural change across the value chain [43,47].

The paper shows that these recommendations, if implemented, will improve the effectiveness of the PAN's RBF programming and increase the sustainability of energy interventions across Nepal. This has implications for other RBF programs as many of the issues shared here are seen across market-based approaches in the International Development sector. If combined with strategic policy making, such as acknowledging the role of stacking in energy use, and embedding the voices of key stakeholders, especially end-users, in the decision-making process, significant progress can be made with poverty-alleviating energy technologies across the globe. Championing a co-produced approach, such as TIME, with all key stakeholder in the technology poverty alleviation value chain is key to achieving SDG7 by 2030.

**Supplementary Materials:** The following are available online at https://www.mdpi.com/article/10.3390/en14102891/s1, Table S1: Reflection Sub-Factor, Table S2: Government Perspective, Table S3: NGO/Business Perspective, Table S4: Co-Ordinating Partner Perspective, Table S5: Community Perspective.

**Author Contributions:** Conceptualization, B.L.R., M.J.C. and S.J.; methodology, B.L.R., M.J.C. and S.J.; data collection, B.L.R.; formal analysis, B.L.R.; writing—original draft preparation, B.L.R.; writing—review and editing, M.J.C. and S.J.; supervision, M.J.C. and S.J. All authors have read and agreed to the published version of the manuscript.

**Funding:** This research was funded by the Rieger Highflyers Scholarship at the University of Nottingham UK.

**Data Availability Statement:** Data available on request.

**Acknowledgments:** Many thanks to all the Practical Action Nepal project staff both in Kathmandu and in the field, especially Min Bikram Malla and Pooja Sharma as well as the tireless efforts of Pratik Bhandari and Subina Shrestha.

**Conflicts of Interest:** The authors declare no conflict of interest. The funders had no role in the design of the study; in the collection, analyses, or interpretation of data; in the writing of the manuscript, or in the decision to publish the results.

**Research Ethics:** All subjects gave their informed consent for inclusion before they participated in the study. The study was approved by the Ethics Committee at the University of Nottingham.

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
