# Peer review of "TIME to Change: An Evaluation of Practical Action Nepal’s Results Based Finance Program"

_energies, doi:10.3390/en14102891_

Round 1

Reviewer 1 Report

The paper aims at analyzing barriers, enablers and engagement strategies for the adoption and sustained use of Improved Cookstoves in Nepal. Introduction and literature review are well written and frame correctly the contribution in current relevant research on the topic.

Data collection and processing are correctly described. Coding text data might be tricky if conduced by multiple people. As each member of the research team might have interpreted differently the documents under investigation. If that is the case,  an intercoder agreement analysis should be performed using for instance the Krippendorff's Alpha indicator. If that is the case I would suggest to address such issue. 

Conclusion & recommendations are interesting; however, I think should be expanded a while for better reporting the results achieved from interviews. I would suggest also to comment them directly with the relevant literature in this field.
There are some minor issues to address related to references that must be adjusted in text e.g line 149 "Error! Reference source not found.." 

Author Response

Thank you for reviewing our paper and for the positive comments. We have corrected the reference error.

We have clarified that only one researcher (the lead author) coded the data to ensure that the interpretation was consistent across the study 9see line 221-223 and 642-644)

We have expanded the conclusion section and made stronger links to the literature throughout the paper. However, as the paper already has a lot of content, we have tried to streamline the additions as much as possible.

Reviewer 2 Report

The paper is lacking a strong scientific hypothesis. I think in general, the article value can be improved. This article generally presented the political implications. This makes the abstract and the conclusion weak and lacks impressive statements. The major editing/revision required are listed below:

  1. The abstract of the paper is not informative and concise. It does not represent the whole article well. Please summarize the main findings of the study.
  2. The introduction part is not reasonable, especially the introduction to the background and meaning of the research is not detailed and adequate enough. I recommend two section – Introduction and Literature review.  There needs to be a literature review, preferably as a separate section and certainly prior to the empirical section(s), situating the research within what is already known relevant to the topic and showing how the specific research question is justified in light of what is and is not yet known or understood. Subject literature should be updated. The new recent references can then be discussed in the text to improve the review quality (literature review and discussion sections). Describe the research limitations.
  3. Conclusions should go deeper, it would be more interesting if the authors focus more on the significance of their findings regarding the importance of the interrelationship between the obtained results and sustainable development, technology development, and policy and management studies.
  4. Very poor editing (Error! Reference source 149 not found, etc.).

Author Response

Thank you for reviewing our article and giving such critical feedback. In regard to the hypothesis issue, hypotheses are not appropriate for qualitative research methods but there are appropriate and clear aim/objectives stated in lines 82-88 with the novelty outlined in lines 90-99. We have addressed the following revisions:

  1. We have checked the abstract and adjusted it accordingly.
  2. As per the journal guidelines we have not include a literature review section, the journal manuscript sections are: “Author Information, Abstract, Keywords, Introduction, Materials & Methods, Results, Conclusions, Figures and Tables with Captions, Funding Information, Author Contributions, Conflict of Interest and other Ethics Statements” (https://www.mdpi.com/journal/energies/instructions). Instead, we have chosen to embed the literature throughout the paper. In addition, the literature surrounding this paper is very limited and all papers of significance are included. The research limitations are described in section 3.3.
  3. We have expanded the conclusion and further linked the recommendations to the reference literature. We have also strengthened the links to the literature throughout the paper. However, as the paper already has a lot of content we have tried to streamline the additions as much as possible.
  4. We apologise for the typos and have fully proofread the article.

Round 2

Reviewer 2 Report

The contribution of this paper to scientific knowledge is good. 

I consider that the article has been modified and significantly improved. 

Author Response

Thank you for reviewing our article.